# *Sphingomonas sediminicola* Is an Endosymbiotic Bacterium Able to Induce the Formation of Root Nodules in Pea (*Pisum sativum* L.) and to Enhance Plant Biomass Production

**DOI:** 10.3390/microorganisms11010199

**Published:** 2023-01-12

**Authors:** Candice Mazoyon, Bertrand Hirel, Audrey Pecourt, Manuella Catterou, Laurent Gutierrez, Vivien Sarazin, Fréderic Dubois, Jérôme Duclercq

**Affiliations:** 1Unité Ecologie et Dynamique des Systèmes Anthropisés (EDYSAN, UMR7058 CNRS), Université de Picardie Jules Verne (UPJV), 80000 Amiens, France; 2Unité Mixte de Recherche 1318 INRA-AgroParisTech, Institut Jean-Pierre Bourgin, Institut National de la Recherche Agronomique et de l'Environnement (INRAE), 78026 Versailles, France; 3Centre de Ressources Régionales en Biologie Moléculaire (CRRBM), Université de Picardie Jules Verne (UPJV), 80000 Amiens, France; 4AgroStation, 68700 Aspach-le-Bas, France

**Keywords:** *Sphingomonas sediminicola*, *Pisum sativum*, PGRP, rhizobacteria, nodulation

## Abstract

The application of bacterial bio-inputs is a very attractive alternative to the use of mineral fertilisers. In ploughed soils including a crop rotation pea, we observed an enrichment of bacterial communities with *Sphingomonas* (*S*.) *sediminicola*. Inoculation experiments, cytological studies, and *de novo* sequencing were used to investigate the beneficial role of *S. sediminicola* in pea. *S. sediminicola* is able to colonise pea plants and establish a symbiotic association that promotes plant biomass production. Sequencing of the *S. sediminicola* genome revealed the existence of genes involved in secretion systems, Nod factor synthesis, and nitrogenase activity. Light and electron microscopic observations allowed us to refine the different steps involved in the establishment of the symbiotic association, including the formation of infection threads, the entry of the bacteria into the root cells, and the development of differentiated bacteroids in root nodules. These results, together with phylogenetic analysis, demonstrated that *S. sediminicola* is a non-rhizobia that has the potential to develop a beneficial symbiotic association with a legume. Such a symbiotic association could be a promising alternative for the development of more sustainable agricultural practices, especially under reduced N fertilisation conditions.

## 1. Introduction

In the last fifty years, conventional agriculture has been based on mechanisation and extensive use of synthetic fertilisers and pesticides to increase yields [1], which had a negative impact on the environment [2]. More sustainable approaches are sought to compete these intensive practices. The approaches that focus on better management practices [3], precision-agriculture technologies [4], plant breeding strategies [5], or taking advantage of crop biodiversity [6] are the most suitable.

In particular, agriculture based on biological interactions, such as those with rhizospheric microorganisms, is very promising [7]. In recent years, it has been shown that inoculation with symbiotic or nonsymbiotic bacterial strains can significantly improve crop productivity [8,9,10,11]. These bacteria are generally described as plant-growth-promoting rhizobacteria (PGPR), which consist of free-living rhizobacteria and symbiotic bacteria. Free-living rhizobacteria such as *Acetobacter*, *Azospirillum*, *Bacillus*, *Pseudomonas*, and *Klebsiella* can improve plant performance by producing growth regulators (e.g., auxin, cytokinins, gibberellin) or through their aminocyclopropane carboxylate deaminase (ACCD) activity, which lowers ethylene levels known to inhibit plant growth [8,10,11]. Other species, such as *Bacillus drentensis* or *Pseudomonas putida* [10,12], can also improve access to nutrients such as nitrogen (N) or phosphorus, or through iron scavenging from the rhizosphere.

In legumes, specific symbiotic bacteria, called rhizobia, are able to develop specialised organs (nodules), in which the symbiont fixes atmospheric N_2_ [13] and converts it into ammonia (NH_3_) that can be readily assimilated by the plant to support its growth and development [14]. In return, the bacteria benefit from the carbohydrates provided by the plants and are protected in a niche that allows N_2_ fixation. Such a symbiosis provides about 200 million tons of fixed N per year [12] and saves 35–60% of the fossil energy needed to produce mineral fertilisers used in conventional agriculture [15]. So far, nodulation of legumes has been described only with α-, β-, and γ-rhizobia [16,17,18]. All the α-rhizobia group within only 15 genera in the order *Hyphomicrobiales* (=*Rhizobiales*): *Rhizobium*, *Mesorhizobium*, *Ensifer* (e.g., *Sinorhizobium*), *Bradyrhizobium*, *Allorhizobium*, *Pararhizobium*, *Neorhizobium*, *Aminobacter*, *Phyllobacterium*, *Microrhizobium*, *Azorhizobium*, *Ochrobactrum*, *Methylobacterium*, *Devosia*, and *Shinellai*, whereas the β-rhizobia are found in three genera of the *Burkholderiales*: *Paraburkholderia*, *Cupriavidus* (e.g., *Ralstonia*), and *Trinickia* [16,17,18]. The γ-rhizobia refers to certain strains of *Pseudomonas* that may be able to induce root nodulation in black locust (*Robinia pseudoacacia*). However, this result remains controversial and requires further confirmation [16].

The process of nodulation that occurs in the legumes/rhizobia symbiotic association is well described [19]. In pea for example, this symbiotic association leads to the formation of indeterminate nodules consisting of a meristematic zone (I), an infection zone (II), an interzone (II–III) characterised by an accumulation of amyloplasts where N_2_ fixation occurs (III), and a senescent zone (IV). One of the key factors for the effectiveness of the symbiosis lies in the differentiation of the bacterium into bacteroids. Bacteroid differentiation is represented by three different morphotypes, namely, the U (undifferentiated), S (large spherical), and E (large elongated bacteroids) [20,21,22,23]. Moreover, the bacteroid morphotype is largely dependent on the host plant rather than its bacterial partner. Therefore, in a biodiversity-based agricultural system, it is important to use the most appropriate legume that will allow soil N_2_-fixing bacterial communities to develop nodules containing the most differentiated bacteroid morphotypes and thus the greatest N_2_-fixing efficiency [23].

Conventional intensive agriculture generally has a negative effect on the diversity of bacterial communities in the soil [24,25,26], especially to the detriment of rhizobia species [27,28]. This shift in soil bacterial communities’ structure is often accompanied by an increase in the abundance of some bacterial groups, such as *Sphingomonas*, particularly during crop rotation, depending on ploughing practices or introduction of cover crops, and also as a consequence of N fertilisation [29,30,31,32]. *Sphingomonas* species are α-*Proteobacteria* commonly found in the rhizosphere and phyllosphere of plants [33,34,35,36,37]. Although *S. melonis* is a plant pathogen [38], most *Sphingomonas* species are beneficial to plants, notably by limiting the development of plant pathogens [39], stimulating plant growth [40], or improving plant resistance to abiotic stresses [41,42]. In a recent study, we found that *S. sediminicola* can represent up to 40% of the total bacterial population in conventionally ploughed soils that contained a crop rotation with pea [29]. To date, studies that have addressed *S. sediminicola* [43,44] have relied on the taxonomy of the bacterium and have not examined its interaction with plants.

In the present study, pea plants were inoculated with *S. sediminicola* to evaluate the beneficial role of this species in pea plant inoculation. Controlled growth chamber experiment, cytological studies, and *de novo* sequencing of the *S. sediminicola* genome showed that *S. sediminicola* is able to colonise pea plants and develop root nodules, allowing the establishment of a symbiotic association that promotes plant biomass production. This finding suggests that *S. sediminicola* has a strong potential to increase both N nutrition and yield in legumes, enhancing their ability to produce green manure during crop rotation in the context of sustainable agricultural production.

## 2. Materials and Methods

### 2.1. Bacterial Materials

The bacterial strains used in this study were *Sphingomonas sediminicola* (DSM-18106) [43], kanamycin-resistant DH5-α *Escherichia coli* carrying GusA under the constitutive promoter p*Neo* in the p*OPS0385* plasmid (Addgene ID #133235) [44], and *Rhizobium leguminosarum* bv. *viciae Rl*v3841 [45]. *S. sediminicola* was grown in R2A medium (VWR, Fontenay-sous-Bois, France) for 72 h at 30 °C (constant shaking at 150 rpm), *R. leguminosarum* at 28 °C (150 rpm) in tryptone-yeast extract (TY) medium [46], and the *E. coli* strain was grown at 37 °C for 24 h (200 rpm) in LB medium (Luria–Bertani, Sigma Aldrich, St. Louis, MO, USA). Bacterial concentration (CFU mL^−1^) was determined by the spiral method using an EasySpiral^®^ automatic plater (Intersciences, Mourjou, France) and counted using a Scan^®^500 (Interscience). At DO_600 nm_ = 0.75, the *S. sediminicola* culture corresponded to 2 10^6^ CFU mL^−1^, while 1 10^8^ CFU mL^−1^ was obtained with *R. leguminosarum* culture.

Mutation of *S. sediminicola* conferring a resistance to rifampicin was generated spontaneously by plating the bacteria on R2A medium containing rifampicin (25 mg L^−1^) [46]. Using bacterial conjugation [47], the plasmid p*OPS0385* carrying GusA was transferred into *S. sediminicola*^Rif^, giving *S. sediminicola*^Rif^ [p*OPS0385*]. This strain was grown in R2A containing 20 mg L^−1^ kanamycin and 25 mg L^−1^ rifampicin.

### 2.2. Controlled Growth Chamber Experiment

Pea (*Pisum sativum*, cv. Douce Provence, Jardiland, Paris, France) seeds were surface-sterilised with a 3.5% (*v*/*v*) bleach solution, cold-stratified for 48 h, and germinated in the dark on a 1% (*w*/*v*) agar medium at 21 °C. Five days after germination, etiolated seedlings were transferred to a Quickpot QP 6 T/20 system (Puteaux, Limas, Villefranche-sur-Saône, France) containing 500 g sterile N-rich (Floradur B, NPK = 18-10-20; Puteaux), N-limited (Horticole Fal, NPK = 4-4-4; Puteaux) potting soil or 200 g sterile N-free substrate (vermiculite; Puteaux). Plants were grown with a 16 h light/8 h dark photoperiod at 21 °C with a light intensity of 400 µmol m^−2^ s^−1^ and inoculated with 1 mL by seed of *S. sediminicola* or *R. leguminosarum* (2 10^6^ and 1 10^8^ CFU, respectively). For noninoculated seeds, 1 mL of R2A or TY bacterial growth medium was used. Plants were grown until flowering (approximately 30 days post-inoculation, dpi) and watered every 3 to 4 days with sterile tap water (N-rich condition), distilled water (N-deprived condition), or N-free Fahraeus solution [48]. The experiment was repeated at least three times independently, each time with 10 to 20 per modality.

### 2.3. Molecular Analyses

From a pool of 3 to 5 surface-sterilised nodules, DNA was extracted using CTAB buffer (2% *w*/*v* cetyltrimethylammonium bromide, 100 mM Tris-HCl, 1.4 M NaCl, 20 mM EDTA). Five nanograms of DNA were used to perform a quantitative polymerase chain reaction (qPCR) with a LightCyler 480 system (Roche Diagnostics, Rotkreuz, Switzerland) using *Sphingomonas*- and *Rhizobium*-specific primers and universal 16SrRNA primers (Appendix A). Calibration ranges (0 to 4 ng µL^−1^) were performed for the 16SrRNA and *Rhizobium* markers from *Rl*v3841, while that for *Sphingomonas* was determined using *S. sediminicola* DNA. The 16S rRNA abundance was corrected to account for variations in gene copy-number according to the 16S copy number database rrnDB [49], i.e., 3 copies for *R. leguminosarum* and 1 copy for *S. sediminicola*.

*S. sediminicola* genome sequencing was performed by Beijing Genomics Institute (BGI, TaI Po, Hong Kong). *S. sediminicola* DNA from a single colony was isolated using the EZNA Tissue DNA Kit (Omega Biotek, Norcross, GA, USA). A negative control consisting only of DNA extraction solutions was included in the DNA extraction procedure to exclude cross-contamination. Genomic libraries were prepared with the Nextera XT Library Prep kit (Illumina, MA, Canton, OH, USA) and checked with the Agilent 2100 Bioanalyzer (Agilent Technologies, Santa Clara, CA, USA) using the Agilent High Sensitivity DNA Kit (Agilent Technologies). Libraries were then pooled in equimolar amounts according to quantification by the Qubit ds-DNA HS Assay (Thermo Fisher Scientific, Waltham, MA, USA) and sequenced using a Hiseq 2500 (Illumina). The quality of the sequencing raw data was checked using FastQC (v0.11.8). Trimmomatic (v0.39) [50] was used to remove the adapters (ILLUMINACLIP option) and low-quality regions (HEADCROP:12 and SLIDINGWINDOW:5:30). Sequences with a length shorter than 80 base pairs (bp) were discarded. A total of 5 independent sequences were generated with an average of 592 Mb. Using the tools of the KBase platform [51], the assembly of the sequences was performed with SPAdes (v3.15.3). Basic assembly properties were assessed using QUAST (v4.4) and genome annotation using Prokka (v1.14.5). Annotations were also manually checked using the BLASTx program from the NCBI database (https://blast.ncbi.nlm.nih.gov/Blast.cgi (accessed on 7 December 2022)) and validated according to sequence similarity scores, E-values, and coverage. Prokka analysis also generated the functional categories of these annotations by comparing them in the SEED system ontology. Gene Ontology analysis and Metacyc pathway analysis were also performed using the QuickGO and MetaCyc databases, respectively [52,53]. Circular visualisations were created using the Circular Genome Visualization Tool (v0.0.2) from the KBase platform.

The similarity of *S. sediminicola* genome sequences compared to that of other bacteria was analysed by calculating the pairwise Average Nucleotide Identity (ANIb and ANIm) values using JSpecies software with the integrated BLAST algorithm [54]. In silico DNA–DNA hybridisations (DDH) values were obtained using the Genome-to-Genome Distance Calculator (GGDC 2.1; http://ggdc.dsmz.de/distcalc2.php (accessed on 7 December 2022), [55]) and the recommended BLAST + alignment. The bacterial genome sequences were downloaded from the GenBank database (https://www.ncbi.nlm.nih.gov/genbank/ (accessed on 7 December 2022)).

### 2.4. Phylogenetic Analyses

Multiple sequence alignments of the *nifH*, *parA*, *nodABCD*, and *nfeD* sequences were performed individually using CLUSTALW (https://www.genome.jp/toolsbin/clustalw (accessed on 7 December 2022)). CLUSTALW first calculated a pairwise genetic distance between the sequences with degrees of similarity between each pair. Then, a phylogenetic tree was constructed for each alignment using the neighbour-joining algorithm with no distance corrections. The trees were generated in the Interactive Tree of Life (iTOL) programs (v6.5.2) [56]. The sequence accessions from NCBI (https://www.ncbi.nlm.nih.gov/ (accessed on 7 December 2022)) are indicated in the phylogenetic tree after the species name. The bootstrap values of the phylogenetic tree are indicated in blue.

### 2.5. Acetylene Reduction Assay

Acetylene (C_2_H_2_) was prepared according to Postgate [57] from 1 g of calcium carbide (CaC_2_; Sigma Aldrich) dissolved in 150 mL of distilled water producing 15.625 mmol of C_2_H_2_. ARA was performed in 250 mL flasks containing detached nodulated root system. Ten percent of the flask volume was replaced with C_2_H_2_ [58], and after 1 h of incubation, a gas aliquot (1 mL) was analysed using an ethylene (C_2_H_4_) analyser (F-950, Felix Instruments, Camas, WA, USA). Ammoniacal silver nitrate (10 g L^−1^) was used to precipitate residual acetylene [59]. Nase activity from the detached root system was expressed in µmol C_2_H_4_ h^−1^ plant^−1^.

### 2.6. Macroscopic and Microscopic Observations

To monitor root nodule colonisation, *S. sediminicola*^Rif^ [p*OPS0385*] (2 10^6^ CFU) was inoculated into 5-day-old etiolated peas growing in sterile vermiculite. Plants were grown under the same conditions as in the controlled growth chamber experiment and watered with N-free Fahraeus solution. The experiment was repeated three times independently. Detection of β-glucuronidase (GUS) enzyme activity (blue colour) in nodule sections of pea inoculated with *S. sediminicola*^Rif^ [p*OPS0385*] was visualised with 5-bromo-4-chloro-3-indolyl glucuronide (X-Gluc; Sigma-Aldrich). The staining buffer was prepared under high stringency conditions for ferro- and ferricyanide concentrations (2.5 mM ferroferricyanide buffer, K_3_Fe(CN)_6_, K_4_Fe(CN)_6_, Sigma-Aldrich) to limit diffusion of GUS product in sodium phosphate buffer (0.1 M NaH_2_PO_4_, 0.1 M Na_2_HPO_4_, pH 7.0 and 0.1% (*v*/*v*) Tritton; Sigma-Aldrich) with X-Gluc (1 mg mL^−1^) previously dissolved in dimethylformamide (Sigma-Aldrich). Staining for GUS activity was performed on surface-sterilised nodules for 45 min at 37 °C in the dark. Macroscopic visualisation of nodules was performed using a stereomicroscope (ZEISS SteREO Discovery V20, Carl Zeiss, Göttingen, Germany). For the visualisation of GUS activity at the intracellular level, detached nodules were fixed for 1 h in 3% (*w*/*v*) glutaraldehyde in 0.1 M phosphate buffer (pH 7.4); dehydrated gradually using ethanol solutions from 15, 30, 50, 70, 80, 90, and 100%; and then embedded in LR white resin (Sigma-Aldrich). Polymerisation was carried out in gelatin capsules for 10 h at 54 °C. Sections of 4 to 6 µm were cut with a histological diamond knife and floated on drops of sterile water on silanised slides at 60 °C. Sections were back stained with periodic acid–Schiff (PAS) [60] to visualise the insoluble polysaccharides (cell wall and starch). For both light and electron microscopy, nodules were fixed in 3% (*w*/*v*) glutaraldehyde dissolved in 0.1 M cacodylate buffer for 3 h at room temperature. Samples were then washed in the cacodylate buffer and post-fixed for 1 h in 1% (*w*/*v*) OsO_4_. The samples were then processed and embedded in resin as described above to visualise the GUS activity. For light microscopy, 1 µm thin sections were placed on silanised slides and stained first with PAS and then with Azur II (Agar Scientific, Stansted, Essex, England) 1% (*w*/*v*) in water. Observations were carried out on a Nikon Eclipse E800 microscope (Nikon, Tokyo, Japan). For electron microscopy, 70 nm ultrathin sections were mounted on 400-mesh nickel grids or 200-mesh nickel carbon formvar-coated grids, dried at 37 °C, counterstained with 5% uranyl acetate in water, and observed with a Tecnai F20 FEI Corpelectron microscope at 120 kV (FEI, Hillsboro, OR, USA).

### 2.7. Statistical Analysis

Means were compared between plants inoculated with *S. sediminicola* or *R. leguminosarum* and non-inoculated plants using the Kruskal–Wallis test (*p* < 0.05) followed by pairwise Wilcoxon rank sum tests with Holm’s *p*-adjust method for multiple comparisons. All statistical testing was carried out in R (v4.0.4; http://www.r-project.org/) using *agricolae* [61] and *ggpubr* [62] packages.

## 3. Results

### 3.1. Sphingomonas sediminicola Improved Pea Biomass Production

First, the effect of *S. sediminicola* inoculation on the main phenotypic traits of pea plants was investigated. On a sterile NPK 18-10-20 potting soil (N-rich substrate), we observed that both shoot and root dry weights of 30-dpi-old plants inoculated with *S. sediminicola* were at least twice higher those of non-inoculated plants (*p* < 0.001; Figure 1a,b). Although shoot height was not altered in the inoculated plants (Figure 1c), we observed that the main root was significantly longer compared to that of non-inoculated plants (*p* < 0.001; Figure 1d).

To compare the effect of *S. sediminicola* inoculation on the different phenotypic traits measured, a rhizobia species (*R. leguminosarum*) that develops a typical N_2_-fixing association in pea was used as a positive control. We observed an increase in both shoot and root biomass production when plants were inoculated with *Rhizobium* (*p* = 0.027 and <0.001, respectively, Figure 1a,b) compared to non-inoculated plants. We also observed that shoot height was higher in plants inoculated with *R. leguminosarum* (*p* < 0.001), while root length was not changed. On a sterile NPK 4-4-4 potting soil (N-limited substrate) and on sterile vermiculite (N-free substrate), an increase in both plant biomass production and shoot height was observed in the presence of *R. leguminosarum*. In all non-inoculated plants (>100), the presence of nodules was never observed. In the plants inoculated with *R. leguminosarum*, the percentage of plants with nodules increased under N-deficient conditions and reached 70% under N-free conditions with 5 to 88 nodules in nodulated plants (Figure 1e).

When plants growing on the N-rich substrate were inoculated with *S. sediminicola*, root nodules were never observed. However, 37.5% and 76.5% of the plants inoculated with *S. sediminicola* and grown under N-limited and N-free conditions, respectively (Figure 1e), developed numerous nodule-like structures similar to those observed when the plants were inoculated with *R. leguminosarum* (Figure 2a). The *Sphingomonas*-specific gene was detected in these detached nodules. In contrast, no amplification was observed when *Rhizobium*-specific primers were used. Conversely, in nodules that developed after inoculation with *R. leguminosarum*, PCR amplification products were detected with the *Rhizobium*-specific primers but not with the *Sphingomonas*-specific primers (Figure 2b). In the detached root system of plants inoculated with *S. sediminicola*, nitrogenase activity was detected at levels comparable to those measured in plants inoculated with *R. leguminosarum* (Figure 2c).

### 3.2. Sphingomonas sediminicola Genome Contains the Genetic Information Necessary to Induce Nodulation

*de novo* whole-genome sequencing showed that *S. sediminicola* had a chromosome of 2,756,125 bp with a GC content of 62.88%. Thus, the genome coverage of the different sequencing runs ranged from 165 to 260X. The bacterial genome was composed of 2764 complete coding sequences (CDSs) and 64 tRNA sequences (Figure 3a). The similarity of *S. sediminicola* genome sequence compared to that of *S. sediminicola* KACC 15039, *Sphingomonas azotifigens*, and some rhizobia was analysed using ANI-Blast (ANIb), ANI-MUMmer (ANIm), and in silico DDH. High ANIb, ANim, and DDH values (>96%) were found between the genomes of *S. sediminicola* and *S. sediminicola* KACC 15039, whereas comparisons with the other selected genomes gave much lower values depending on the three types of analysis (Table 1).

Among these CDSs, the most abundant were those encoding proteins involved in amino acids and related metabolism (6.8%), membrane transport (5.8%), protein fate (5.3%), carbohydrate metabolism (4.3%), and some small molecules (including cofactors, vitamins, prosthetic groups) (4.2%) (Figure 3b). More precisely, and despite the detection of 1018 CDSs encoding proteins of unknown function, we found that some show strong homology with genes involved in iron uptake (*fur*, *piuB*), storage (*bfrA*, *bfrD*, *bfd*), transport (*feoB*, *fecR*, *fieF*, *tonB*), scavenging (*entS*, *yfiZ*), phosphorus transport (*pstABCS*), and signal transduction (*phoBHRU*). Genes encoding proteins involved in tryptophan biosynthesis (*trpABCD*) and, interestingly, in N_2_ fixation (*fixJKL*, *nifSU-like)* were also identified (Figure 3a, Appendix A).

The presence of a plasmid of 519,958 bp was also detected. It contained a number of CDSs showing strong similarity to genes involved in the replication of low-copy-number plasmids (*parABC*), nodule formation and development (*nodABCDEFLMPQ*, *nolKVT*; *noeLJ*), N_2_ fixation (*nifBHRUX*, *fixGHIJNOPQS*), and nitrate assimilation (*narGHIJKQ*, *napA*, *nirBD*). In addition, this structure harboured two clusters of CDSs sharing strong similarities with genes of type three (T_3_SS, *escRSTUV; sctDOPQ*) and four (T_4_SS, *virB2*,*3*,*4*,*6*,*9*,*10*,*11*) encoding proteins involved in secretion systems. We also identified other CDSs sharing strong homology with *bacA* (*Bacteroid development protein A*), *acdS genes*, and a number of genes involved in the regulation of bacterial motility when chemical attraction occurs in the rhizosphere (*cheCY*, *fliL*, *flbD*, *flgDF*) (Figure 3c, Appendix A).

Phylogenetic analysis showed that the *nifH* sequence of *S. sediminicola* clustered with that of another *Sphingomonas* species (*S. azotifigens*) in the clade that included *Azorhizobium caulinodans*. In addition, we found that *S. sediminicola nifH* sequence was also closer to those of the selected β-rhizobia and *Bradyrhizobium* species compared with the sequences of other α-rhizobia (Figure 4). Conversely, the partitioning system (*parA)* sequences of *S. sediminicola* and *S. azotifigens* were more closely related to that of *Rhizobium* than that of other *Sphingomonas* species (Appendix A). Unlike those of *nodC*, the *nodA* and *nodB* sequences of *S. sediminicola* were found to be quite similar to those of *Rhizobium* (Appendix A). The *nodD* sequence of *S. sediminicola* was close to that of *Azorhizobium caulinodans*, whereas the *nfeD* sequences formed a cluster unique to *Sphingomonas* species and distinctly different from those of other bacterial genera (Appendix A).

### 3.3. Sphingomonas sediminicola Induced the Formation of Nodules

To unequivocally demonstrate that *S. sediminicola* is responsible in pea plants for nodule formation, inoculation of pea plants with a modified *S. sediminicola* strain containing a GUS reporter gene (*S. sediminicola^Rif^* [p*OPS0385*]) was performed in sterile vermiculite and with an N-free nutrient solution. This experiment was performed to determine if the reporter gene activity could be detected in the nodule tissues. Compared with the wild-type strain, the changes in shoot and root biomass production and in the number of nodules per plant were similar (Appendix A). We also verified that the wild-type strain did not exhibit GUS activity, either under free-living conditions or in the nodules of inoculated plants. In mature nodules (30 dpi), GUS activity (blue staining) was detected only in the central area of the nodules (Figure 5a, arrow), whereas roots (R), vascular bundles (VB), cortical cell layers (C), and apical area (AA) remained unstained. Thinner nodule sections confirmed that GUS activity was present in the majority of cells in the central zone of the nodule (Figure 5b–d). In this area, we also observed that some cells containing large vacuoles remained unstained (stars) (Figure 5b,c), similar to those of the cortical area (C) (Figure 5c). PAS counterstaining also revealed the presence of numerous starch granules (SG) appearing as dark-red dots. At higher magnification (Figure 5d), GUS staining was visible in small particles corresponding to bacteroids (B) with an expected size of 1 µm.

To refine the cellular structure of the central zone of the nodules and to determine whether the infected cells contained bacteroids, light and electron microscopy experiments were performed on thin and ultrathin sections of nodule tissue, respectively. An overview of a nodule longitudinal thin section (Figure 6a) confirmed that three distinct cell types were clearly observed, consisting of an apical area (AA) characteristic of indeterminate pea nodules [61] containing small cells without starch, an intermediate area (IA) composed of larger blue-stained cells without starch, and a fully differentiated and mature area (MA) represented by large blue-stained cells with numerous starch granules (Figure 6a). Infection threads (ITs) were frequently observed in the intermediate area. These ITs contained small spherical or rod-shaped, blue-stained particles resembling bacteria (Figure 6b). These blue-stained particles were also detected in the cytoplasm of the adjacent cells (Figure 6b, arrows). Magnification of the central area allowed us to detect within the cell cytoplasm the presence of many almost unstained ovoid particles (Figure 6c) surrounded by a white halo (Figure 6d, arrows). It is likely that a membrane surrounding these particles was present, thus indicating that they could be bacterial symbiosomes.

Transmission electron microscopy (TEM) experiments confirmed that the particles observed by light microscopy were symbiotic bacteria and that they were polymorphic. In the intra- or intercellular IT near the apical area visible in Figure 6b, the average surface of the bacteria (B) was 0.46 ± 0.11 µm^2^ (Figure 6e and Appendix A). We also observed that the bacteria were surrounded by an electron-transparent matrix (arrows), which was itself included in a slightly denser fibrous matrix with no separation of membranes between the two matrices. In many cells of the apical area and in cells containing IT, the size of the bacteria ranged from 0.5 to 1 μm, and as in the central zone of the nodule, we observed many membrane structures, suggesting that an endocytosis process had occurred (Figure 6f,g, arrows). This process probably resulted in the formation of symbiosomes called bacteroids, each consisting of a single symbiotic bacterium and a peribacteroid membrane (Figure 6h, arrow). The size of this structure changed while the nodule developed to reach an average surface of 1.62 ± 0.66 µm^2^ in the intermediate area and 4.37 ± 1.56 µm^2^ in the mature area (Appendix A and Figure 6i).

## 4. Discussion

Enrichment of *Sphingomonas* has been frequently observed in agricultural soils after crop rotation, ploughing, cover crops establishment, and in relation to N fertilisation [29,63]. However, these studies dealt with metabarcoding characterisation of bacterial communities and did not directly address the function of *Sphingomonas* species [64,65,66]. In particular, the predominance of *S. sediminicola* in the pea rhizosphere raised the question of its role as a biological indicator in conventionally tilled agricultural soils [29]. In the present study, we showed that *S. sediminicola* improved plant biomass production similarly to other PGPR [8,11,67], such as *Azospirillum brasilense* [68], *Bacillus licheniformis* [69], *Pseudomonas aeruginosa* [70], or other *Sphingomonas* species. For example, *S. LK11* was able to stimulate growth and biomass production in soybean [71] and tomato [40]. A similar effect was observed in *Arabidopsis thaliana* with *S. Cra20* [64]. The mechanisms underlying these beneficial effects mainly involve facilitation of nutrient acquisition (phosphorus, iron), modulation of plant hormone levels and ACCD activity [72].

The *S. sediminicola* genome sequence allowed us to predict the potential capabilities of this strain. First, the size of the *S. sediminicola* genome (2.75 Mb) is among the smallest compared to other *Sphingomonas* genomes, ranging from 2.88 Mb in *Sphingomonas* sp. *W1-2-3* [66] to 6.58 Mb in *S. sanxanigenens* [73]. As ANI values below 95–96% or 70% for DDH are the cut-off points for delineating bacterial species, the genome of *S. sediminicola* analysed in the present study was found to be quite different from that of rhizobia species and to be similar to that of *Sphingomonas sediminicola* KACC 15,039 [43]. However, we identified an extrachromosomal element of 520 kb containing a CDS that has strong homologies with low-copy-number plasmids carrying a partitioning locus, called the ParABC cassette [74]. Therefore, this extrachromosomal element could be a plasmid that we named p*Ss01*. Within the *Sphingomonas* genus, many plasmids have been identified, and most of them have this ABC partitioning system [75,76] and are also large, such as p*CAR3* in *Sphingomonas* sp. KA, p*ISP0* in *Sphingomonas* sp. MM-1, and p*SWIT01* in *Sphingomonas wittichii* RW1 [77]. We found some CDSs on the chromosome of *S. sediminicola* that show homology with genes involved in iron absorption (*fur*), iron storage (bacterioferritins *bfrA* and ferrodoxin *bfrD*), and iron transport (ferrous iron transporters, *feoAB*), as well as a gene encoding a pump involved in ferrous ion efflux (*fieF*) [78]. In parallel, we identified CDSs close to genes related to the synthesis of enterobactin siderophores (*entS*) and siderophore transporters (*yfiZ, tonB*). These genes are present in many PGPR such as *Enterobacter* sp. J49 [79], *Bradyrhizobium yuanmingense* [80], or *R. cellulosilyticum* [81] and *Sphingomonas pokkalii* [82]. CDSs involved in phosphorus metabolism, such as specific inorganic phosphorus transporters (*pstABCS*) and transcriptional regulators of phosphorus metabolism (*phoBHRU*), have also been identified on the chromosome of *S. sediminicola*. These components are found in several PGPRs, such as *Burkholderia cenocepacia* [83], *Paenibacillus sonchi* [84], or *Pseudomonas psychrotolerans* [85]. CDSs sharing homologies with genes related to tryptophan biosynthesis (*trpABCD*), which may act as a biosynthetic precursor of auxins, were found in the *S. sediminicola* genome. Many PGPR, such as *Acetobacter, Azospirillum, Bacillus, Bradyrhizobium, Burkholderia, Klebsiella, Pseudomonas, Rhizobium, Xanthomonas* [86,87], *and Sphingomonas LK11* [37,66], are auxin producers. Interestingly, we identified a CDS in p*Ss01* that shows strong homology with *acdS*, a gene encoding the enzyme ACCD. This bacterial enzyme catalyses the reduction and cleavage of the ethylene precursor (ACC) produced by plants [88]. Therefore, the next step will be to further characterise these bacterial properties that may explain the improved plant performance when *S. sediminicola* is inoculated not only with pea, but also with other monocotyledonous and dicotyledonous species.

One of the most striking results of our study was that *S. sediminicola* was able to induce nodulation on pea roots. Root nodule development is a process restricted to very specific plant–bacteria interactions, usually involving rhizobia and legumes [16,17,18]. During this interaction, and under N-deficient conditions, flavonoids are excreted by the plant into the rhizosphere and interact with the bacterial transcription factor NodD [89], which activates the transcription of a set of nodulation (*nod*) genes. In the well-characterised *Rhizobium*–legume symbiosis, *nodABC* transcription leads to the synthesis and secretion of lipochitooligosaccharides, the so-called Nod factor (NF), which triggers the entry of the bacteria and nodulation [89,90]. Interestingly, we found several CDSs with strong similarity to *nodABC* and *nodD* in plasmid *pSs01*. The presence or absence of specific nodulation genes (*nod*, *noe*, and *nol*) determines NF structure [91] by controlling host specificity [17]. The NF structure is also specific to the host plant species and can also be altered according to environmental conditions [90]. In *R. leguminosarum*, *nodEL* are involved in the formation of root nodules in pea and clover [92,93], while in *R. meliloti, nodHPQ* are required for alfalfa nodulation. It is also likely that *nodHPQ* are involved in the process of soybean nodulation in *Bradyrhizobium japonicum* [93]. A number of CDSs identified in the genome of *S. sediminicola* shared strong homology with genes controlling host specificity, such as *nodELMPQ* [92,94,95]. Such a finding suggests that *S. sediminicola* could be able to induce nodule formation in temperate legume species other than pea, as well as in tropical legumes.

It is known that other biological components such as root colonisation and symbiont recognition and suppression of the plant immune system are involved in the establishment of a rhizobia–legume symbiosis. All these steps require specific secretion systems, namely, types 1, 3, and 4 [96], which translocate effectors into the host plant. In *R. leguminosarum bv. viciae*, the type 1 protein secretion system (T_1_SS), which is encoded by the *prsD* and *prsE* genes, is involved in the secretion of EPS-glycanases [97]. These enzymes play a key role in biofilm formation both during root colonisation and in the initial steps of symbiotic interaction [96]. In the chromosome of *S. sediminicola*, CDSs encode proteins similar to *prsD* and *prsE*, suggesting that EPS-glycanases can be excreted by the bacteria.

Host-produced flavonoids induce expression of the *nodD* gene encoding NF, which also activates expression of type 3 secretion genes (T_3_SS). We identified in the plasmid p*Ss01* that many CDSs share strong similarities with the members of the T_3_SS [98] that form a cluster of genes similar to that found in the T_3_SS core components of the Rhc-I subgroup [99]. T_3_SS is the only secretion system involved in the establishment of symbiosis with legumes [100]. This type of T_3_SS is present in many rhizobia, such as *R. elti*, *Ensifer* (*Sinorhizobium*) *fredii*, and *Bradyrhizobium japonicum* [99]. Close to the T_3_SS cluster in p*Ss01*, we also identified nine CDSs sharing strong similarities with T_4_SS–b, which is functionally similar to the T_3_SS-Rch-1 [100]. T_4_SS has been identified in some rhizobia, such as *Mesorhizobium loti* R7A [101] and *R. etli* CFN42 [102].

In temperate legumes that develop indeterminate nodules, N_2_ fixation occurs in infected cells located in the zone III. This zone is visible in the root nodules of pea plants inoculated with *S. sediminicola* (Figure 5). It is known that the leghaemoglobin in this zone prevents the inhibition of the NAse enzyme by O_2_ while maintaining cellular respiration [103,104]. Thus, ATP can be produced via oxidative phosphorylation by a cytochrome c oxidase (Cbb3-type) in the mitochondria [104,105,106]. The genes *fixNOQP* [16,105], which encode cytochrome c oxidase, are located near the *fixGHIS* genes that enable the assembly of Cbb3. We found two clusters of CDSs in the *S. sediminicola* genome that have very strong homologies with *fixNOQP* and *fixGHIS*. These two clusters were located next to each other on p*Ss01*. They were also found in a similar position as in the symbiotic plasmids p*SymA* and p*110* of *Ensifer* (*Sinorhizobium*) *meliloti* [17] and *R. leguminosarum*, respectively [105], and in the symbiotic islands of *Bradyrhizobium japonicum* USDA 110 [105]. The regulation of the *fixNOQP* and *fixGHIS* operons is usually controlled by the *fixJKL* genes, which were also present in *pSs01*. As in some rhizobia, such as *R. leguminosarum* 3841 [105], the *fixKL* regulatory genes were located on the chromosome of *S. sediminicola*, while the *fixJ* gene was found on the plasmid p*Ss01*. As in *Ensifer meliloti* [105], *fixNOQP* were also located in *S. sediminicola* next to the CDSs, exhibiting strong homology to genes encoding proteins involved in the control of N metabolism, such as *napA* (a periplasmic nitrate reductase), *nirDB*, and *narGHIJKQ* (involved in nitrite and nitrate reduction, respectively).

Regulation of O_2_ partial pressure during symbiotic N_2_ fixation by rhizobia is also an essential process for maintaining a fully active Nase. In these bacterial species, the *nifAB* and *nifHDKEN* genes are involved in the synthesis and assembly of the enzyme [17,107]. *nifA* encodes a transcriptional regulator required for the transcription of *nifB*, *nifN*, and *nifHDKEX*. *nifH* encodes a reductase that enables the production of NH_4_^+^ from atmospheric N_2_, while *nifB* is required for the biosynthesis of a specific cofactor located at the active site of Nase. The assembly of this cofactor, composed of iron molybdenum (FeMo-co), is in turn initiated by the gene products *nifS* and *nifU* [107,108]. Several CDSs with similarity to *nifBHU* were identified in the megaplasmid of *S. sediminicola*. The *nifS* gene was located on its chromosome as well as other CDSs sharing homologies with *nifX*, *nifE*, and *nifN*, which encode the NAse FeMo-cofactor [16]. Such a distribution of the different *nif* genes on a plasmid and on the chromosome has also been found in other rhizobia, such as *Ensifer meliloti* [17]. The analysis of the *S. sediminicola* genome allowed the identification of several genes encoding proteins involved in the onset of atmospheric N_2_ fixation. However, some of the genes essential to this process, such as *nifD* and *nifK* [109], were not identified in this species. As this analysis also revealed the presence of several genes encoding proteins of unknown function, further work is required to determine whether some of these genes are essential for a fully functional Nase enzyme. Our phylogenetic analysis showed that the acquisition of nodulation genes by *S. sediminicola* involved dozens of genes and was thus rather complex. Therefore, it can be hypothesised that the sequences of *S. sediminicola nifD* and *nifK* are so distant from those of other rhizobia that they could not be identified on the basis of sequence homologies alone.

At the ultrastructural level, different cell types were identified in the mature root nodules of pea plants inoculated with *S. sediminicola*, which are characteristic of indeterminate nodules. From their apical to their basal area, these nodules are composed of an initial zone of meristematic cells that give rise to cells that form the infection zone with some cell layers full of starch granules, followed by cells in which the bacteria progressively differentiate [20,110]. Light and electron microscopic observations allowed us to follow the different steps in the establishment of the symbiotic association at the cellular level, including the penetration of the bacteria into the root cells and the formation of a symbiosome. In particular, the presence of IT and an electron-translucid matrix probably corresponding to an infection droplet that allows a direct contact between the bacteria and the root cell membrane was clearly visible in the apical zone. We were also able to visualise the process of bacterial endocytosis through the plasma membrane in the infected zone, followed by the development of a symbiosome compartment in which the bacteria differentiate into bacteroids [111]. This differentiation is known to be plant host dependent [112], and in legumes belonging to the inverted repeat–lacking clade (IRLC), this process leads to terminal differentiation of their bacterial endosymbionts, resulting in endoreduplication of the genome, cell enlargement, and loss of cell division ability [113]. In nodules of pea inoculated with *S. sediminicola*, we identified bacteria in the infection zone that were similar in size to those previously observed in free-living bacteria [43] (i.e., 0.5 µm diameter by 1 µm length), while in zone III, the size of the bacteroids increased fivefold and is characteristic of the differentiation of bacteroids when they begin to fix atmospheric N_2_ [20]. This observation suggests that in the symbiotic association of *S. sedimincola* and pea, the resulting indeterminate nodules contain E-type bacteroids. However, it would be useful to analyse the expression of NCR-like genes by RT-qPCR and the DNA content by flow cytometry.

In conclusion, our results rekindle current concepts on nodulation of legumes both biological and evolutionary perspectives, especially for further agronomic applications. For almost a century, it was generally assumed that all legumes could form nodules only when inoculated with bacteria of the order *Hyphomicrobiales* (=*Rhizobiales*), which belong to the α-*Proteobacteria* (α-rhizobia). Then, in the last two decades, the discovery of modulating bacteria belonging not to the classical α-rhizobia but to the β- or γ-*Proteobacteria* (β- or γ-rhizobia) has overturned the postulate that only α-rhizobia are able to develop N_2_-fixing symbiotic associations with legumes [16,17,18]. By showing that within the α-*Proteobacteria*, nodulation of legumes is not restricted to the *Hyphomicrobiales* but can also occur in the *Sphingomonadales*, it can be suggested that the classification of α-rhizobia taxa could be revised to include *S. sediminicola* or other species belonging to the *Sphingomonadales*.

## 5. Conclusions

In summary, our work demonstrates that *S. sediminicola* possesses strong potential with respect to legume N nutrition, yield, and their ability to produce green manure, as this soil bacterium can develop functional root nodules when inoculated with pea. To further promote the biological and agronomic importance of the *Sphingomonas*–legume association, more research is needed to (1) decipher the structural and regulatory elements involved in such an association both during its establishment and during the onset of N_2_ fixation; (2) evaluate the benefits of such an association in agricultural systems where peas or other legumes are grown either as the main crop, as green manure, or as an associated crop in the presence or absence of *Rhizobium*; and (3) investigate its short- or long-term effects on the microbial community in the rhizosphere of the surrounding plant and on soil properties.

## Figures and Tables

**Figure 1 microorganisms-11-00199-f001:**
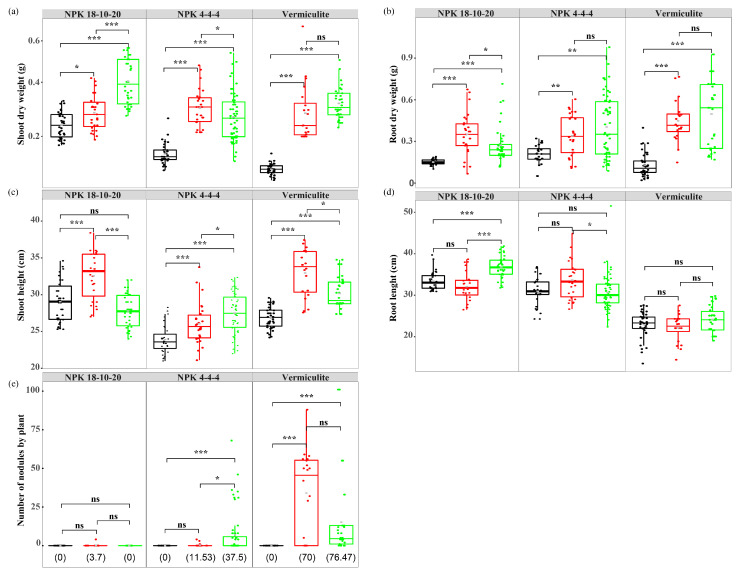
Phenotypic traits of uninoculated peas (black) and peas inoculated with *Rhizobium leguminosarum* (red) or *Sphingomonas sediminicola* (green) grown under N-rich (NPK = 18-10-20), N-limited (NPK = 4-4-4), and N-free (vermiculite) substrate. Plants were harvested at flowering (about 30 days post-inoculation). (**a**) Dry weight of shoots and (**b**) roots. (**c**) Shoot height and (**d**) root length. (**e**) Number of root nodules in plants grown under the three different N-fertilisation conditions. The percentage of nodulated plants is given in brackets. Statistical differences were based on Wilcoxon rank sum tests with Holm’s p-adjust. *, *p* < 0.05; **, *p* < 0.01; ***, *p* < 0.001; ns, not significant.

**Figure 2 microorganisms-11-00199-f002:**
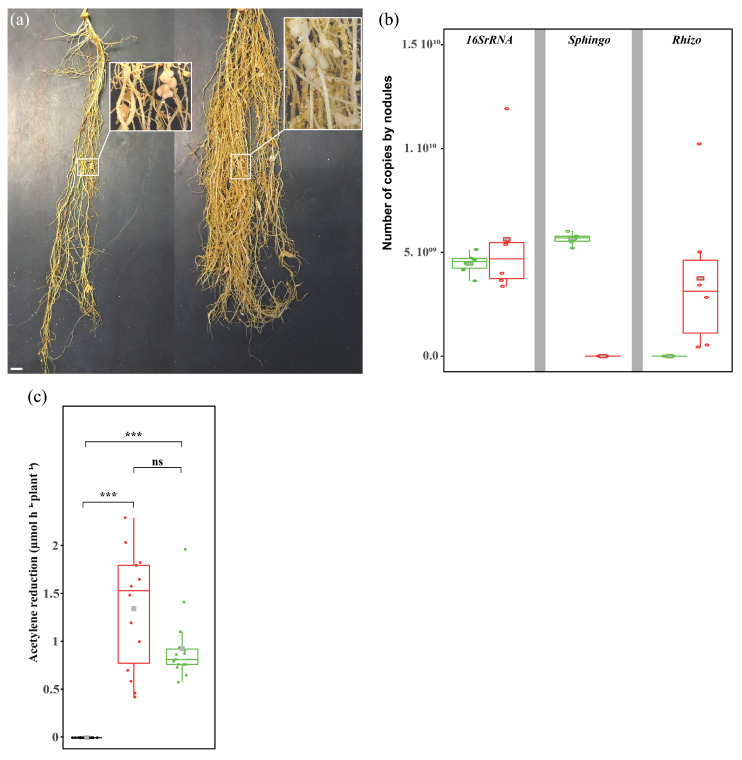
Characteristics of nodules. (**a**) The picture shows the presence of root nodules in plants inoculated with *Rhizobium leguminosarum* (**left**) and *Sphingomonas sediminicola* (**right**) grown under N-limited conditions. Scale = 1 cm. (**b**) Number of copies of the universal 16SrRNA as well as *Sphingomonas* and *Rhizobium* genes in nodules of pea plants inoculated with *S. sediminicola* (green) and *R. leguminosarum* (red). (**c**) Measured nitrogenase activity using the ARA with the noninoculated and non-nodulating root system (black) of pea plants, as well as the nodulated root system of peas inoculated with *R. leguminosarum* (red) and *S. sediminicola* (green). Statistical differences were based on Wilcoxon rank sum tests with Holm’s *p*-adjust. ***, *p* < 0.001; ns, not significant.

**Figure 3 microorganisms-11-00199-f003:**
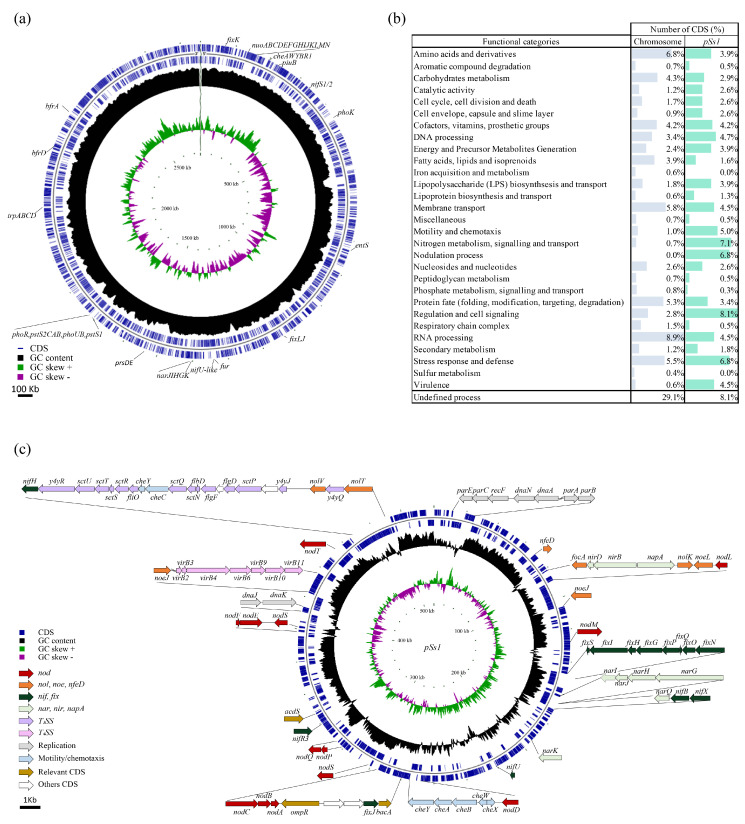
Structure of the *Sphingomonas sediminicola* genome. (**a**) The size of the bacterial chromosome is 2.75 Mb. (**b**) Functional categories of chromosomal and plasmid CDSs according to the SEED system ontology. (**c**) Circular map of the *S. sediminicola* plasmid (*pSs1*). The broad arrows in the outer circle indicate genes encoding *nod* (red), *nol*, *noe*, *nfeD* (orange), *nif*, *fix* (dark green), *nar*, *nir*, *napA* (light green), T3SS (dark pink), T4SS (light pink), replication (grey), motility, chemotaxis (light blue), relevant (brown), and other coding regions (white). In (**a**,**c**), the outer concentric circles (blue) encompass the location and direction of the coding region (CDS), the circle in the middle (black) indicates the % GC content, and the inner circle indicates the GC skew [(G − C)/(G + C)] as positive (green) or negative (purple).

**Figure 4 microorganisms-11-00199-f004:**
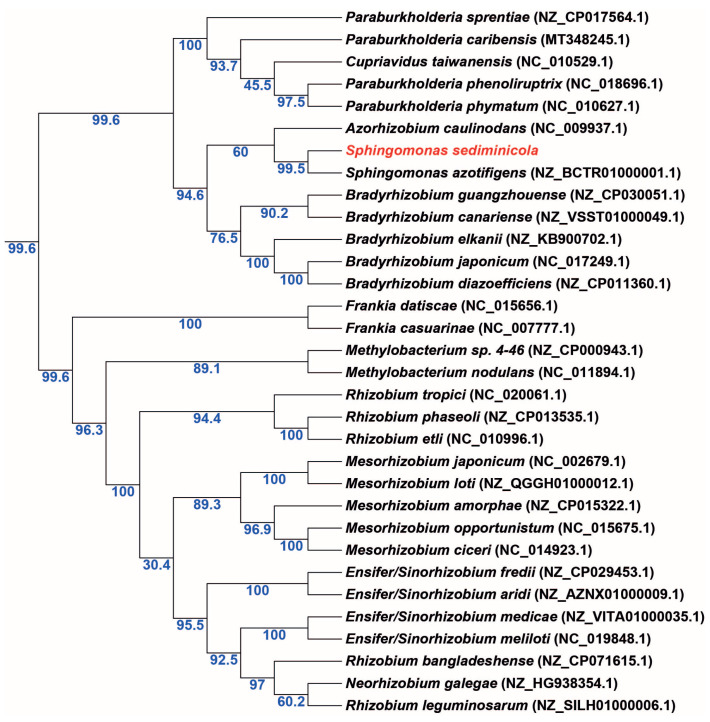
Phylogenetic tree of *nifH* sequences. Sequence accessions from NCBI (https://www.ncbi.nlm.nih.gov/) are indicated after the name of the species. *Sphingomonas sediminicola* is written in red for easy location. Bootstrap values of the phylogenetic tree are indicated in blue font.

**Figure 5 microorganisms-11-00199-f005:**
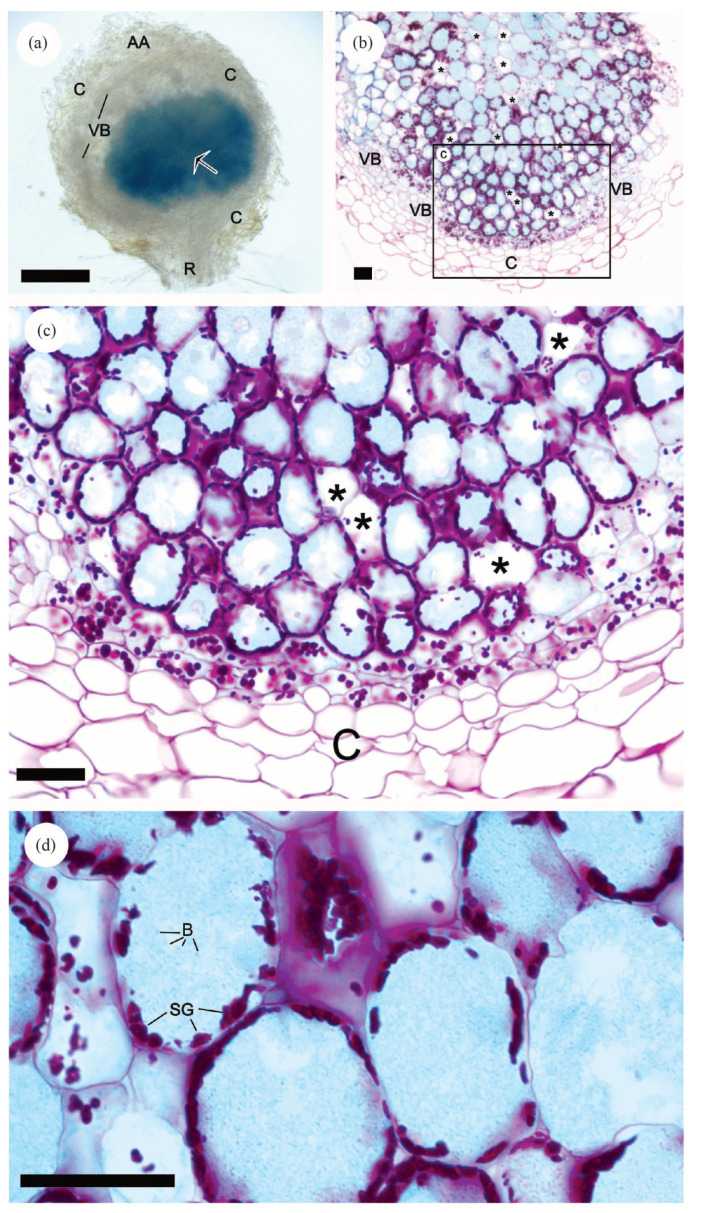
Overview and thin sections of pea nodules after inoculation with *Sphingomonas sediminicola* [pOPS0385] expressing a *GUS* reporter gene. (**a**) View of the nodule in which GUS enzyme activity (blue staining) was detected in the central zone, while the roots (R), apical area (AA), cortical cell layers (C), and vascular bundles (VB) remained unstained. (**b**–**d**) Thin sections of a nodule counterstained with periodic acid–Schiff (PAS) show that GUS activity was only detected in the central area (**b**,**c**). Most of the blue staining was associated with a granular texture (**d**) compatible with the GUS signal inside bacteria (B), while no signal was seen in the cortical cell layers (C), the vascular bundle (VB), nor in some cells containing large vacuoles (*). PAS counterstaining revealed the presence of numerous starch granules (SG). Bars represent 500 μm (**a**) or 50 μm (**b**–**d**).

**Figure 6 microorganisms-11-00199-f006:**
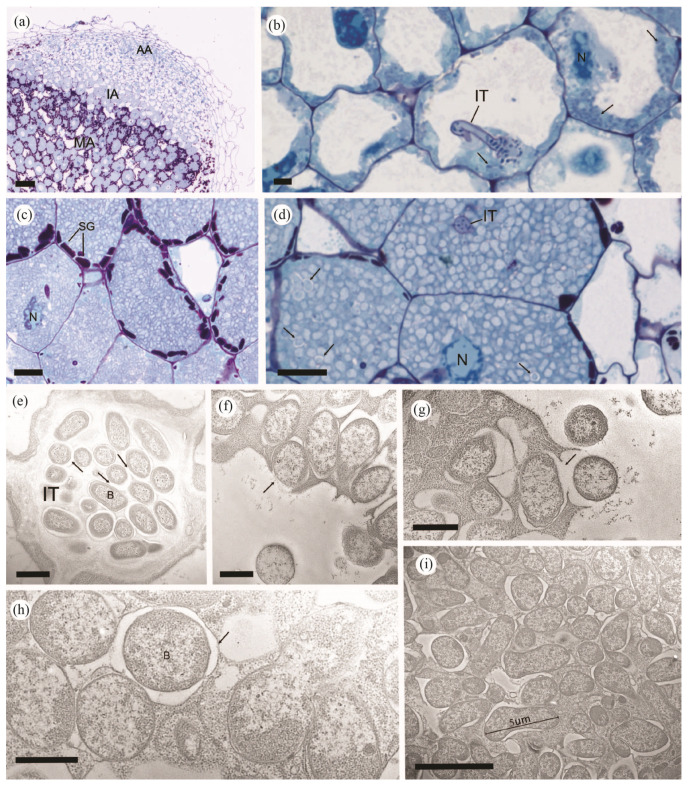
Detailed structural and ultrastructural analysis of a nodule formed after inoculation of peas with *Sphingomonas sediminicola*. (**a**) Light microscopic photograph of a nodule showing different cell types characterising an apical area (AA), an interzone (IA), and a mature area (MA), with the latter two containing symbiosomes. (**b**) Detail of an infection thread (IT) invading the cell and releasing a bacterium in the apical zone. (**c**,**d**) Enlargements of the symbiosomes in the cells of the central zone of the nodule. Arrows underline obvious symbiosome structures (N) indicating cell nuclei and (SG) starch granules. (**e**–**i**) Ultrathin sections of nodules taken with a transmission electron microscope. (**e**) Cross-section of an infection thread (IT) with free-living bacteria (B) inside an electron transparent matrix (arrows). The presence of bacteroids surrounded by a peribacteroid membrane can be seen in (**f**,**g**), together with the appearance of an endocytosis process (arrows). (**h**) Detail of a symbiosome consisting of bacteria (B) clearly isolated from the host cytoplasm by a surrounding membrane (arrow). (**i**) Details of fully enlarged and differentiated symbiosomes in the mature area of the nodules. Bars represent 100 μm (**a**), 10 μm (**b**–**d**), 1 μm (**e**–**h**), and 5 μm (**i**).

**Table 1 microorganisms-11-00199-t001:** Average nucleotide identity (ANI) and in silico DNA–DNA hybridisation (DDH) comparisons between *Sphingomonas sediminicola*, *S. sediminicola* KACC 15039, *Sphingomonas azotifigens*, and some rhizobia genome sequences.

Species	GenBank Assembly Accession	ANIb [%]	ANIm [%]	In Silico DDH (%)
*Sphingomonas sediminicola KACC 15039*	GCA_014489515.1	99.94	99.93	96.50
*Sphingomonas azotifigens NBRC 15497 [T]*	GCA_002091475.1	69.89	83.68	13.20
*Methylobacterium* sp. *4-46*	GCA_000019365.1	67.13	82.82	12.70
*Methylobacterium symbioticum SB0023/3 [T]*	GCA_902141845.1	67.07	83.47	12.60
*Methylobacterium nodulans ORS 2060 [T]*	GCA_000022085.1	66.90	83.63	12.70
*Mesorhizobium opportunistum WSM2075 [T]*	GCA_000176035.2	66.90	83.49	12.60
*Mesorhizobium amorphae CCNWGS0123*	GCA_001686985.1	66.84	83.31	12.60
*Azorhizobium caulinodans ORS 571 [T]*	GCA_000010525.1	66.83	83.83	12.70
*Rhizobium phaseoli R630*	GCA_001664205.1	66.82	83.18	12.70
*Mesorhizobium japonicum MAFF 303099 [T]*	GCA_000009625.1	66.80	82.62	12.70
*Bradyrhizobium elkanii USDA 76 [T]*	GCA_012871055.1	66.75	82.56	12.70
*Bradyrhizobium japonicum USDA 6 [T]*	GCA_013752735.1	66.74	83.21	12.60
*Sinorhizobium fredii NGR234*	GCA_000018545.1	66.69	83.25	12.70
*Mesorhizobium loti DSM 2626 [T]*	GCA_003148495.1	66.67	83.51	12.70
*Rhizobium etli CFN 42 [T]*	GCA_000092045.1	66.66	83.41	12.70
*Sinorhizobium meliloti SM11*	GCA_000218265.1	66.66	83.35	12.70
*Bradyrhizobium diazoefficiens USDA 110 [T]*	GCA_001642675.1	66.62	83.23	12.70
*Mesorhizobium ciceri biovar biserrulae WSM1271*	GCA_000185905.1	66.61	83.37	12.70
*Bradyrhizobium canariense BTA-1 [T]*	GCA_019402665.1	66.57	82.94	12.60
*Rhizobium leguminosarum bv. viciae 3841*	GCA_000009265.1	66.57	84.33	12.60
*Rhizobium bangladeshense BLR175 [T]*	GCA_017357245.1	66.50	84.16	12.60
*Neorhizobium galegae bv. officinalis bv. officinalis str. HAMBI 1141*	GCA_000731295.1	66.36	82.98	12.60
*Rhizobium tropici CIAT 899 [T]*	GCA_000330885.1	66.22	84.20	12.60
*Cupriavidus taiwanensis LMG 19424 [T]*	GCA_000069785.1	65.45	82.92	12.60
*Paraburkholderia sprentiae WSM5005 [T]*	GCA_001865575.2	64.85	83.05	12.60
*Paraburkholderia phymatum STM815 [T]*	GCA_000020045.1	64.82	83.07	12.60
*Frankia alni ACN14a [T]*	GCA_000058485.1	64.73	82.75	12.50
*Paraburkholderia phenoliruptrix BR3459a*	GCA_000300095.1	64.71	83.92	12.50
*Frankia casuarinae CcI3 [T]*	GCA_000013345.1	64.23	82.42	12.50

## Data Availability

Whole-genome sequencing data associated with this study has been deposited in the NCBI Sequence Read Archive under accession number PRJNA818132.

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
