# Peer review of "Sphingomonas sediminicola Is an Endosymbiotic Bacterium Able to Induce the Formation of Root Nodules in Pea (Pisum sativum L.) and to Enhance Plant Biomass Production"

_microorganisms, 2023, doi:10.3390/microorganisms11010199_

Round 1
Reviewer 1 Report
The study presented by Mazoyon et al, describes the symbiotic association between Pisum sativum and the non-rhizobial species Sphingomonas sediminicola. The authors clearly demonstrate through different approaches that S. sediminicola is able to induce nodule formation in pea. Additionally, by genome sequencing they show that S. sediminicola potentially possess nod genes. The manuscript is well presented and written, with detailed information of the protocols employed. The study is indeed interesting, since contributes to the discovery of microorganisms that enhance plant growth. However, the work can be significantly improved by exploring other angles of this symbiotic association, both in the writing of the manuscript and in the incorporation of additional data.
Major Comments
The discovery of a non-rhizobial microorganism capable to induce nodule formation and perform nitrogen fixation is one of the relevant findings of this study. However, the authors poorly address this finding in the introduction and discussion.
Additionally, the authors could exploit their discovery, by exploring the capacity of S. sediminicola to induce nodule formation in other legumes, for instance, Medicago truncatula or Lotus japonicus. This experiment would unveil if the symbiotic capacity of S. sediminicola is restricted to P. sativum or extended to other legumes.
Pisum sativum belongs to the IRLC, where terminal differentiation of bacteroids is apparently a distinctive feature. The authors state that the size of the symbiotic bacteria (S. sediminicola) is increased in the infection zone and the nodules of P. sativum contained E-type bacteroids (line 575-579). This would be interesting to explore but the authors do not show any data regarding this feature. The authors could compare the size of the free-living bacteria, vs the isolated bacteroids, as previous studied have done in different IRLC species.
Minor:
Line 64, paragraph needs reference(s)
Line 65-73. More appropriate references (articles and reviews) can be found in the literature for the statements regarding terminal bacteroid differentiation and bacteroids morphotype.
Figure 1. Indicate the harvesting timepoint in the legend of the figure.
Throughout the manuscript the authors state that temperate legumes form indeterminate nodules but this is not a general rule.
Line 575-579 The authors did not show any data regarding terminal differentiation of bacteroids.
Author Response
The discovery of a non-rhizobial microorganism capable to induce nodule formation and perform nitrogen fixation is one of the relevant findings of this study. However, the authors poorly address this finding in the introduction and discussion.
This important point has been more strongly emphasized at the end of the introduction and discussion.
Additionally, the authors could exploit their discovery, by exploring the capacity of S. sediminicola to induce nodule formation in other legumes, for instance, Medicago truncatula or Lotus japonicus. This experiment would unveil if the symbiotic capacity of S. sediminicola is restricted to P. sativum or extended to other legumes.
Preliminary studies allowed to show that S. sediminicola can induce nodule formation both in clover and alfalfa, but not in soybean. However, these data cannot be included in the present study as they need to be refined and conducted on a larger scale for reliable statistical and biological analyses.
Pisum sativum belongs to the IRLC, where terminal differentiation of bacteroids is apparently a distinctive feature. The authors state that the size of the symbiotic bacteria (S. sediminicola) is increased in the infection zone and the nodules of P. sativum contained E-type bacteroids (line 575-579). This would be interesting to explore but the authors do not show any data regarding this feature. The authors could compare the size of the free-living bacteria, vs the isolated bacteroids, as previous studied have done in different IRLC species.
Following the reviewer’s recommendation, this point was mentioned at the end of the discussion.
Minor:
Line 64, paragraph needs reference(s)
Line 65-73. More appropriate references (articles and reviews) can be found in the literature for the statements regarding terminal bacteroid differentiation and bacteroids morphotype.
The additions and modifications have included in the revised version of the paper.
Figure 1. Indicate the harvesting timepoint in the legend of the figure.
The indication of the harvesting timepoint has been included in the legend of figure 1.
Throughout the manuscript the authors state that temperate legumes form indeterminate nodules but this is not a general rule.
This point has been taken into account in the revised version of the paper.
Line 575-579 The authors did not show any data regarding terminal differentiation of bacteroids.
In the results section we have indicated that there is an increase in the size of the bacteroids (up to a 5-fold). This is also included in the discussion.

Reviewer 2 Report
This paper describes the symbiotic properties of Sphingomonas sediminicola (Ss) bacteria in association with pea. The bacteria can form root nodules displaying nitrogen fixation activity. The authors characterized the beneficial impact of Ss inoculation on pea growth. They measured the symbiotic nitrogen fixation activity. They sequenced the bacterial genome and analyzed the phylogeny using few symbiotic markers. They characterized the structure of the pea-Ss nodules. Globally the experiments described are properly done and fit to the current standards of the scientific community. These findings are new because all of the known pea symbionts enabling nitrogen fixation belong or are closely related to the Rhizobium leguminosarum complex species.
To my point of view, this paper deserves to be published in Microorganisms and will improve our knowledge of pea beneficial interactions. However, I have several comments that may be addressed by the authors in a revised version of their manuscript before acceptance:
- - More information is required concerning the sequencing data and the genome assembly strategy used to finally get only two replicons without any gap in the sequence (numbers of scaffolds, sequencing coverage, strategy).
- - I was not able to find any sequence data in NCBI with the PRJNA818132 ID. I didn’t find any genome corresponding to the work in the NCBI database.
- - At the genome era, proper phylogenetic analysis requires to analyze at the genome level the distance between Ss genome and other bacterial genomes present in databases. This allows to define bacterial genospecies. The most popular way to do it is to use the average nucleotide identity (ANI). Several web tools may be used by the authors, this one for example https://jspecies.ribohost.com/jspeciesws/. Alternatively, although less popular, GGDC method may be used instead of ANI https://www.dsmz.de/services/online-tools/genome-to-genome-distance-calculator-ggdc . The analysis will allow the author to unambiguously distinguish Ss from Rhizobium leguminosarum and others rhizobial complex species. In addition to global genome analysis similar comparison based on chromosome only or plasmid only may be used to detect obvious effect of plasmid related horizontal transfer.
- - Phylogeny based on individual molecular markers remains interesting and deserves to be maintained in the paper but didn't allow any general conclusion. I suggest to do the phylogeny using the bac120 set of 120 universal bacterial core genes used by Parks et al (2017; https://doi.org/10.1038/s41564-017-0012-7) or Young et al (2021; https://doi.org/10.3390/genes12010111).
- - Author have monitored plant growth parameter, nodule numbers, as well as nitrogen fixation activity (per plant) but do not give information on nodule biomass (biomass/nodule, total nodule biomass per plant), that may be more relevant for functional analysis than the nodule number per plant.
- - The pictures of bacterial colonies doesn’t give any clear information on phenotype. I suggest deleting the sentence of line 270-273, as well as the (b) panel of the Fig.2.
- - In the introduction, the sentences of Lines 69 to 83 sound as an overinterpretation of the literature. Although in few species a correlation between N2 fixation efficiency, bacteroid morphotypes and differentiation may be observed, the genericity of this statement remains speculative. I would suggest reducing this part as this topic is not directly relevant to the study.
Author Response
This paper describes the symbiotic properties of Sphingomonas sediminicola (Ss) bacteria in association with pea. The bacteria can form root nodules displaying nitrogen fixation activity. The authors characterized the beneficial impact of Ss inoculation on pea growth. They measured the symbiotic nitrogen fixation activity. They sequenced the bacterial genome and analyzed the phylogeny using few symbiotic markers. They characterized the structure of the pea-Ss nodules. Globally the experiments described are properly done and fit to the current standards of the scientific community. These findings are new because all of the known pea symbionts enabling nitrogen fixation belong or are closely related to the Rhizobium leguminosarum complex species.
To my point of view, this paper deserves to be published in Microorganisms and will improve our knowledge of pea beneficial interactions. However, I have several comments that may be addressed by the authors in a revised version of their manuscript before acceptance:
- More information is required concerning the sequencing data and the genome assembly strategy used to finally get only two replicons without any gap in the sequence (numbers of scaffolds, sequencing coverage, strategy).
More details have been provided in the sections Materials and Methods and Results.
- I was not able to find any sequence data in NCBI with the PRJNA818132 ID. I didn’t find any genome corresponding to the work in the NCBI database.
The information can be found in NCBI, but we agree that the search was unsuccessful because in its release it is indicated “upon publication”, which probably temporally limits its access.
Here is the link to the data:
https://dataview.ncbi.nlm.nih.gov/object/PRJNA818132?reviewer=aoo8lehjd97tpmup6mr7u7s46v
- At the genome era, proper phylogenetic analysis requires to analyze at the genome level the distance between Ss genome and other bacterial genomes present in databases. This allows to define bacterial genospecies. The most popular way to do it is to use the average nucleotide identity (ANI). Several web tools may be used by the authors, this one for example https://jspecies.ribohost.com/jspeciesws/. Alternatively, although less popular, GGDC method may be used instead of ANI https://www.dsmz.de/services/online-tools/genome-to-genome-distance-calculator-ggdc . The analysis will allow the author to unambiguously distinguish Ss from Rhizobium leguminosarumand others rhizobial complex species. In addition to global genome analysis similar comparison based on chromosome only or plasmid only may be used to detect obvious effect of plasmid related horizontal transfer.
The authors thank the reviewer for the suggestion and for providing the corresponding methods of analysis. This allowed us to analyze both the average nucleotide identity (ANI) and the genome-to-genome distance. The results of these analyzes have been included in the Results section and presented in Table 1.
- Phylogeny based on individual molecular markers remains interesting and deserves to be maintained in the paper but didn't allow any general conclusion. I suggest to do the phylogeny using the bac120 set of 120 universal bacterial core genes used by Parks et al (2017; https://doi.org/10.1038/s41564-017-0012-7) or Young et al (2021; https://doi.org/10.3390/genes12010111).
We agree that on a phylogenic point of view such analysis could allow to provide further understanding concerning the ability of S. sediminicola to develop root nodules in pea. However, we were unable to have access to the sequences of the bac120 set of 120 universal bacterial core genes.
- Author have monitored plant growth parameter, nodule numbers, as well as nitrogen fixation activity (per plant) but do not give information on nodule biomass (biomass/nodule, total nodule biomass per plant), that may be more relevant for functional analysis than the nodule number per plant. Jérôme?
The authors agree that this information can provide a better understanding of the functionality of the nodules. Unfortunately, these parameters were not measured.
The pictures of bacterial colonies doesn’t give any clear information on phenotype. I suggest deleting the sentence of line 270-273, as well as the (b) panel of the Fig.2.
The sentence both in the text and in Figure 2b has been deleted. The Materials and Methods section corresponding to Figure 2b has also been removed.
In the introduction, the sentences of Lines 69 to 83 sound as an overinterpretation of the literature. Although in few species a correlation between N2 fixation efficiency, bacteroid morphotypes and differentiation may be observed, the genericity of this statement remains speculative. I would suggest reducing this part as this topic is not directly relevant to the study.
This part of the Introduction has been reduced.

Round 2
Reviewer 1 Report
I found unsatisfactory the revised version of the study, since the major comments were very partially addressed. The changes were hard to track within the revised study since these were not marked or the lines/paragraphs were not precisely indicated in their response.
I couldn´t find any additional information in the introduction and discussion regarding non-rhizobial species able to induce nodule formation. As I mentioned in my previous revision round, this is a relevant finding that must be discussed.
In their response, the authors state that preliminary information indicates that S. sediminicola is able to nodulate other legumes but they need to confirm it. Such analysis is not very challenging and in a few weeks such information could be obtained.
An important feature of IRLC legumes is their capacity to impose terminal differentiation onto the bacteroids. Once again, the comparison of the size between the free-living bacteria and bacteroids is not a challenging task, however, the authors did not perform such analysis and only presented the measurement of a single bacteroid in a TEM micrograph.
